# Molecular Basis of HER2-Targeted Therapy for HER2-Positive Colorectal Cancer

**DOI:** 10.3390/cancers15010183

**Published:** 2022-12-28

**Authors:** Ayumu Yoshikawa, Yoshiaki Nakamura

**Affiliations:** 1Department of Gastroenterology and Gastrointestinal Oncology, National Cancer Center Hospital East, Kashiwa 277-0882, Japan; 2International Research Promotion Office, National Cancer Center Hospital East, Kashiwa 277-0882, Japan; 3Translational Research Support Section, National Cancer Center Hospital East, Kashiwa 277-0882, Japan

**Keywords:** colorectal cancer, ctDNA, *HER2* amplification, liquid biopsy, the TRIUMPH study

## Abstract

**Simple Summary:**

Human epidermal growth factor receptor 2 (*HER2)* amplification has emerged as a biomarker of metastatic colorectal cancer (mCRC). Prospective clinical trials have demonstrated the efficacy of HER2-targeted therapies for HER2-positive mCRC and explored the underlying mechanisms. To improve the therapeutic efficacy of this strategy, many clinical trials with various HER2-targeted agents are ongoing. This review discusses the molecular basis of HER2-targeted therapeutic strategies for patients with HER2-positive mCRC.

**Abstract:**

Human epidermal growth factor receptor 2 (*HER2*) amplification has emerged as a biomarker in colorectal cancer (CRC), occurring in 1–4% of metastatic CRC (mCRC). In addition to conventional methods, such as immunohistochemistry and fluorescence in situ hybridization, next-generation sequencing-based tissue or circulating tumor DNA analysis has recently been used to identify *HER2* amplification and assess HER2 overexpression. Prospective clinical trials have demonstrated the efficacy of HER2-targeted therapies in HER2-positive mCRC. The TRIUMPH study, a phase II study of dual HER2 antibodies, i.e., pertuzumab plus trastuzumab, demonstrated promising efficacy for patients with HER2-positive mCRC confirmed by tissue-and/or blood-based techniques, which led to the regulatory approval of this combination therapy in Japan. The mechanisms associated with efficacy and resistance have also been explored in translational studies that incorporate liquid biopsy in prospective trials. In particular, *HER2* copy number and co-alterations have repeatedly been reported as biomarkers related to efficacy. To improve the therapeutic efficacy of the current strategy, many clinical trials with various HER2-targeted agents are ongoing. This review discusses the molecular basis of HER2-targeted therapeutic strategies for patients with HER2-positive mCRC.

## 1. Introduction

The number of biomarker-stratified therapeutic options for patients with metastatic colorectal cancer (mCRC) has increased as the molecular understanding of colorectal cancer (CRC) has progressed. Human epidermal growth factor receptor 2 (*HER2*/*ERBB2*) amplification has emerged as a biomarker of mCRC and occurs in 1–4% of patients with mCRC [1,2,3,4,5]. To identify *HER2* amplification, in addition to conventional tests, such as immunohistochemistry (IHC) and fluorescence in situ hybridization (FISH), tissue- or plasma-based next-generation sequencing (NGS) is used [6,7]. Resistance to anti-epidermal growth factor receptor (EGFR) therapy and the efficacy of anti-HER2 treatment suggest that *HER2* amplification is an actionable alteration in CRC [8,9,10]. With ongoing research into *HER2*-amplified CRC, new therapeutic strategies have been developed. As there is no preclinical biological signal for HER2 targeting with single-agent trastuzumab or tyrosine kinase inhibitors (TKIs), combination therapies or antibody-drug conjugates (ADCs) have been clinically developed for patients with *HER2*-amplified mCRC [11,12]. Currently, combination of trastuzumab with pertuzumab, or lapatinib, or trastuzumab conjugated to deruxtecan are recommended in the guidelines [13]. In Japan, the combination of pertuzumab and trastuzumab was approved for treating patients with *HER2*-amplified mCRC in 2022 [14,15].

In this review, we provide an outline of the current biomarkers and novel diagnostic methods, including NGS and liquid biopsy, for HER2-positive CRC. Furthermore, we also discuss recent therapeutic approaches that have been developed to target the HER2 pathway in CRC.

## 2. Molecular Characteristics of *HER2*-Amplified CRC

CRC was one of the first solid tumors to be molecularly characterized. Several genes and pathways have been shown to be involved in tumor initiation and growth. A series of recurrent mutations in *APC*, *KRAS*, *SMAD4*, and *TP53* are crucial recurrent driver events that accumulate during adenoma formation and progression to sporadic CRC, often correlating with specific stages of the cancer development process [16,17]. Molecular studies have also shown that alterations in WNT–β-catenin, membrane receptor tyrosine kinases (RTKs), and downstream MAPK and PI3K signaling pathways are nearly ubiquitous events in CRC [17,18]. In 2012, unsupervised clustering analysis of the Cancer Genome Atlas (TCGA) data of on 276 primary CRC cases for somatic copy number (CN), whole-exome sequencing, DNA methylation, messenger RNA and microRNA sequencing, and protein array yielded two subtypes: hypermutated and non-hypermutated tumors [17]. Hypermutated tumors were associated with right-sided tumors and hypermethylation, whereas somatic CN alterations (CNAs) were enriched in non-hypermutated tumors, suggesting chromosomal instability. One of the regions of focal amplification, identified in 4% of CRC cases, involved chromosomal region 17q21.1, which contains *HER2*. An international effort coordinating analytics compared six independent transcriptome-based subtyping systems, resulting in a consensus molecular subtype (CMS) classification that enabled the categorization of most CRCs into one of four CMSs: CMS1 (microsatellite instability immune subtype), CMS2 (canonical subtype), CMS3 (metabolic subtype), and CMS4 (mesenchymal subtype) [19]. The relationship between CMS and biological and clinical features and prognosis has been reproduced in multiple studies [20,21,22]. However, the relationship between *HER2* amplification and CMS remains unclear. In the original manuscript on the development of CMS, the prevalence of *HER2* amplification was 0% for CMS1, 1% for CMS2, 3% for CMS3, and 5% for CMS4, whereas CNAs were frequently observed in CMS2 and CMS4 [19,23]. The relationship between *HER2*-amplified CRC and molecular subtypes, such as CMS, requires further investigation.

HER2 is a member of the epidermal growth factor receptor (HER/EGFR/ERBB), an RTK protein, also including EGFR, HER3, and HER4. Binding of a ligand to the extracellular region of EGFR, HER3, and HER4 results in a three-dimensional conformational change, allowing for dimer formation with other HER family members. However, no endogenous ligands for HER2 are known. It is thought to form a heterodimer by binding to homodimers or activated EGFR, HER3, and HER4 to form a signal transduction molecule. Activation of HER2 signal is triggered by either heterodimerization with another HER protein or homodimerization of HER2 [24]. *HER2* amplification leads to aberrant signaling in downstream pathways, particularly those that result in the activation of extracellular signal-regulated kinase 1/2 (ERK1/2) signaling, similar to *RAS* or *BRAF* V600E mutations in CRC. *HER2* amplification is mainly observed in *RAS*/*BRAF* wild-type mCRC, although approximately 20% of *HER2* amplification co-occurs with *RAS* mutations [5,25,26]. This mechanism of ERK1/2 signaling cascade activation strongly suggests resistance to anti-EGFR therapy for patients with *HER2*-amplified mCRC. Yonesaka et al. found a large region of CN gain on chromosome 17 encompassing *ERBB2* in a cetuximab-resistant CRC cell line [12]. In addition, they showed that, in 233 cetuximab-treated patients with mCRC, the median progression-free survival (PFS) and overall survival (OS) were significantly shorter in patients with *HER2*-amplified mCRC. Resistance to second- or later-line anti-EGFR therapy in *HER2*-amplified mCRC has been replicated in several subsequent studies [8,12]. In contrast, Sartore-Bianchi et al. showed the clinical impact of *HER2* amplification on first-line chemotherapy used in combination with an anti-EGFR monoclonal antibody [10]. This study that compared the PFS of anti-EGFR monoclonal antibody-containing therapy in patients with HER2-positive (n = 76) compared to HER2-negative mCRC (n = 108) showed that the hazard ratio (HR) was 1.52 (95% confidence interval (CI), 0.82–1.52) in the third- or fourth-line treatment, while it was 0.93 (95% CI, 0.59–1.46) in the first-line treatment. In a retrospective analysis of the CALGB/SWOG 80,405 trial, a randomized phase III trial evaluating treatment of mCRC patients with first-line FOLFOX or FOLFIRI in combination with cetuximab or bevacizumab, the OS of patients with *HER2*-amplified mCRC was similar for those receiving cetuximab or bevacizumab [27]. These results suggested that the negative impact of *HER2* amplification on the efficacy of anti-EGFR therapy is stronger in later lines, in which the confounding impact of chemotherapy decreases.

It remains unclear whether *HER2* amplification occurs frequently after anti-EGFR therapy as an acquired resistance mechanism. Bertotti et al. showed that the frequency of *HER2*-positive mCRC was 13.6% in 44 patients with *KRAS* wild-type tumors, who had progressed to cetuximab or panitumumab treatment [11]. This finding suggests that a HER2 test after anti-EGFR therapy could potentially identify acquired *HER2* amplification, which can be targeted by anti-HER2 treatment. However, we previously revealed that, in an evaluation of emerging genomic alterations in circulating tumor DNA (ctDNA) after anti-EGFR therapy in 55 patients with mCRC that was initially wild-type *RAS* and non-*HER2*-amplified, *HER2* amplifications were newly identified after therapy only in three patients (5.5%), with a low median CN (adjusted value = 4.0), suggesting that it could not be targeted by anti-HER2 treatment [28].

## 3. Diagnostic Procedures of *HER2*-Amplified CRC

IHC and FISH are typically used to identify *HER2* amplification in CRC tumors, using modified and customized criteria for assessing HER2 positivity in breast and gastric cancers. In the HERACLES Diagnostic Criteria, with archival test (n = 256) and clinical validation (n = 830) cohorts, an IHC staining scale of 0–3 was retained [7]. However, because the cellularity of *HER2* amplification is quite homogeneous, with all positive cases displaying amplification in >50% of cells, a 3+ HER2 score in more than 50% of tumor cells by IHC or a 2+ HER2 score and a *HER2*/*CEP17* ratio >2 in more than 50% of tumor cells by FISH are required to be considered a HER2-positive diagnosis. An international collaboration between GI-SCREEN (Japan), NCTN-SWOG (USA), and Korea harmonized the diagnostic criteria, integrating data based on IHC and FISH for HER2-positive mCRC [29]. Their assessment of 475 CRC tumor samples showed heterogeneity in HER2 expression, resulting in a 10% cutoff for HER2-positive cells. In addition, HER2-positive CRC cells are usually gland-forming types that show strong lateral membrane staining, and basal membrane staining is not always observed. Thus, complete lateral or circumferential membrane staining is required for HER2 positivity.

NGS allows for the sequencing of a large number of nucleotides in a short time frame, resulting in the simultaneous detection of multiple biomarker alterations. The internal collaborative study among GI-SCREEN, NCTN-SWOG, and Korea demonstrated a strong correlation between CN by FISH and NGS [29]. A study of 102 patients with CRC also showed 92% concordance between IHC and NGS in identifying *HER2*-amplified tumors, which increased to 99% concordance when cases with equivocal result in IHC were considered positive [6]. The increasing number of biomarkers for mCRC, such as *NTRK* fusions, high tumor mutational burden, wild-type *RAS*, *BRAF* V600E mutation, and high microsatellite instability may justify the use of NGS in patients with mCRC, but its cost-effectiveness requires consideration.

Liquid biopsy analysis of ctDNA is another promising method to identify *HER2* amplification in mCRC. In sequencing of ctDNA from 1107 patients with mCRC in the GOZILA study, a large-scale ctDNA genomic profiling program in Japan, *HER2* had the highest median plasma CN (pCN) among all CNAs, suggesting the role of HER2 amplification as a driver and targetable alteration in mCRC [14]. In addition, in 75 patients tested using both ctDNA and tissue HER2 testing, the sensitivity and specificity of *HER2* amplification of ctDNA versus tissue were 82% and 83%, respectively. Patients with *HER2* amplification in tissue, but not in ctDNA, had a significantly lower ctDNA fraction, indicating that low tumor shedding is associated with false-negative ctDNA results. A study analyzing ctDNA in 47 evaluable plasma samples enrolled in the HERACLES trial found *HER2* amplification in 46 samples, yielding a sensitivity of 97.9% for identifying HER2-positive CRC [30]. Since the pCN of *HER2* amplification is generally affected by ctDNA fraction, the pCN adjusted by ctDNA fraction (adjusted pCN: ApCN) may be more useful for assessing the CN. Indeed, ApCN was more strongly correlated with the ISH *HER2*/*CEP17* ratio and *HER2* CN determined by quantitative reverse transcriptase-polymerase chain reaction than was pCN. Taken together, the combination of IHC, FISH, tissue NGS, and ctDNA analysis are methods that can support clinical trial enrollment and treatment decisions.

## 4. Efficacy of HER2-Targeted Treatment for Patients with HER2-Amplified mCRC in Clinical Trials

In HER2-positive tumors, available therapeutic agents targeting HER2 include anti-HER2 antibodies, TKIs, and ADCs (Figure 1). Monoclonal antibodies targeting HER2, such as trastuzumab and pertuzumab, bind to the extracellular domain of HER2 and inhibit dimerization, resulting in antibody-dependent cellular cytotoxic (ADCC) effects. TKIs including lapatinib, pyrotinib, neratinib, and tucatinib inhibit cell proliferation by blocking the tyrosine kinase activity of HER2. Lapatinib, pyrotinib, and neratinib are pan-HER TKIs, but tucatinib is a HER2-selective TKI. Trastuzumab emtansine (T-DM1) and trastuzumab deruxtecan (T-DXd) are HER2-targeted ADCs, which are covalently attached to a microtubule inhibitor and a topoisomerase inhibitor, respectively. When trastuzumab binds to HER2 on the tumor surface, its ADC is internalized and causes cytotoxicity by releasing the cytotoxic agents.

In preclinical studies, anti-HER2 drug monotherapy using anti-HER2 antibody or a pan-HER TKI did not suppress the tumor growth of *HER2*-amplified CRC due to insufficient suppression of HER2/EGFR activation or induction of HER3 phosphorylation [11,31]. However, the combination of trastuzumab and lapatinib potently impaired growth by preventing HER2/EGFR/HER3 reactivation. These preclinical findings suggested the potential of dual targets for HER2 and EGFR/HER3 to overcome the resistance against anti-HER2 antibody or a pan-HER TKI alone.

The HERACLES trial, conducted in Italy, was the first clinical trial of dual HER2-blocakde, which evaluated the efficacy of combined trastuzumab and lapatinib in patients with *KRAS* exon 2 wild-type and *HER2*-amplified mCRC, who were heavily pretreated with standard-of-care therapies, including prior EGFR-targeted antibodies [32]. Of the 914 patients with *KRAS* exon 2 (codons 12 and 13), 48 (5%) had HER2-positive tumors, defined as HER2 IHC3+ or HER2 IHC2+ with a *HER2*/*CEP17* ratio ≥ 2 by FISH. The objective response rate (ORR), or the primary endpoint, was 30% (8/27 patients). In the HERACLES-B trial, which evaluated the efficacy of pertuzumab and T-DM1 in patients with *RAS*/*BRAF* wild-type and *HER2*-amplified mCRC refractory to standard treatments, the ORR was 9.7% and the median PFS was 4.1 months (95% CI, 3.6–5.9 months) [33].

The efficacy of pertuzumab and trastuzumab has also been evaluated in other clinical trials, including MyPathway, TAPUR, and TRIUMPH [14,34,35]. The MyPathway was a phase IIa, multiple basket study designed to evaluate the activity of established targeted therapies for non-approved indications in the USA, based on the tumor molecular profile, including pertuzumab plus trastuzumab for *HER2*-amplified solid tumors [34]. In the *HER2*-amplified mCRC cohort, the ORR, or the primary endpoint, was 32% (18/57). In the TAPUR trial, a basket trial conducted in the USA that aimed to describe the safety and efficacy of commercially available targeted anticancer drugs prescribed for patients with advanced cancer with a potentially actionable genomic variant, the ORR of patients with *HER2*-amplified mCRC treated with pertuzumab and trastuzumab was 14% (4/28) [35]. The TRIUMPH trial was a Japanese phase II trial of pertuzumab and trastuzumab, seeking to identify patients with *HER2*-amplified mCRC prospectively, by ctDNA genotyping in addition to conventional tissue HER2 testing by IHC and FISH (Table 1) [14]. The TRIUMPH trial enrolled 30 patients with *RAS* wild-type and *HER2*-amplified mCRC, including 27 patients who were confirmed as HER2-positive by tissue testing and 25 who were confirmed by ctDNA genotyping, of which 22 overlapped. The primary endpoint was ORR, analyzed for the two primary populations: tissue- and ctDNA-based HER2-positive results. In the TRIUMPH trial, real-world clinical outcomes for patients with *RAS* wild-type and *HER2*-amplified mCRC treated with non-HER2-targeted standard-of-care therapies were also assessed as a reference using the SCRUM-Japan Registry in this observational cohort study of real-world data of patients with advanced solid tumors [15]. The study met the primary endpoint with a confirmed ORR of 30% in 27 tissue-positive patients and 28% in 25 ctDNA-positive patients, as compared to an ORR of 0% in a matched real-world reference population treated with standard-of-care salvage therapy. The median duration of response was 12.1 months (95% CI, 2.8 months to not reached) in patients with tissue positivity and 8.1 months (95% CI, 2.8 months to not reached) in patients with ctDNA positivity. Thus, the results of the TRIUMPH trial indicated that patients with *HER2*-amplified mCRC identified by ctDNA genotyping benefited from dual-HER2 blockade, similar to HER-2-positive patients identified by conventional tissue analysis.

Tucatinib is an oral TKI that is highly selective for HER2. The efficacy of trastuzumab and tucatinib in patients with *RAS* wild-type and *HER2*-amplified mCRC was evaluated in the MOUNTAINEER trial. At a median follow-up of 20.7 months, the confirmed ORR among 84 patients who received the combination treatment was 38.1%, with a median PFS of 8.2 months (95% CI, 20.3–36.7 months) [36]. In addition to the dual HER2-blockade, the activity of T-DXd in *HER2*-amplified mCRC was also explored in the DESTINY-CRC01 trial [37]. Of the 53 patients with *RAS*/*BRAF* wild-type and HER2-positive mCRC enrolled in the trial, 24 (45.3%) patients had a confirmed objective response.

## 5. Resistance Mechanism against HER2-Targeted Treatment in *HER2*-Amplified mCRC

Biomarkers associated with the efficacy and resistance of HER2-targeted treatment have been evaluated in previous clinical trials. A higher ORR in HER2 IHC 3+ mCRC than in HER2 IHC 2+ and ISH-positive mCRC cases was shown in the HERACLES and DESTINY-CRC01 trials [32,37]. Indeed, the CN of *HER2* and concurrent genomic alterations have been repeatedly reported as biomarkers in several trials (Table 2). An exploratory analysis of the HERACLES trial showed that a *HER2* CN cutoff of 9.45 could stratify responders versus non-responders, with a median PFS of 29 weeks and 16 weeks in patients with CN above and below this cutoff, respectively (HR 0.67, 95% CI 0.6–0.8) [32]. Furthermore, patients with ApCN ≥ 25.82 had a significantly longer PFS than those with ApCN < 25.82 (median, 22.5 vs. 14.8 weeks; *p* = 0.0347) [30]. The TRIUMPH trial demonstrated that the *HER2* CN determined by tissue NGS and ctDNA ApCN correlated significantly with the clinical benefits of pertuzumab and trastuzumab [14]. Superior efficacy in patients with a high ApCN was also shown for treatment with T-DXd in the DESTINY-CRC01 study [38].

Concurrent genomic alterations have also been suggested to be associated with resistance to anti-HER2 therapy in *HER2*-amplified mCRC. In the MyPathway study, the tumor response was achieved in only 1 of 13 (8%) patients with *KRAS*-mutant mCRC, while the ORR was 40% in *KRAS* wild-type mCRC [34]. Although cases with *KRAS* mutations were excluded from the HERACLES trial, baseline plasma ctDNA analysis identified *RAS* mutations, and the three patients with *RAS* mutations with high clonality showed no tumor responses to anti-HER2 therapy with trastuzumab and lapatinib [30].

The association between baseline tissue and ctDNA genomic alterations and the efficacy of pertuzumab and trastuzumab was assessed in the TRIUMPH trial. Interestingly, co-alterations of oncogenes, such as *HER2* and *BRAF*, were detected in only a small number of non-responders based on archival tissue NGS, whereas baseline ctDNA genotyping showed that amplifications or clonal mutations of genes related to RTK/RAS and PI3K signaling pathway were markedly enriched in non-responders (Figure 1) [14]. In the stratification by concurrent alterations and *HER2* CN, patients with a *HER2* CN above the threshold and no concurrent oncogenic alterations showed significantly better PFS (median in those with versus those without favorable factors, tissue-based, 6.2 versus 2.2 months HR = 0.28 (95% CI, 0.11–0.74); ctDNA-based: 5.6 versus 1.6 months, HR = 0.14 (95% CI, 0.05–0.39)) and OS (median, tissue-based, 23.4 versus 7.4 months, HR = 0.17 (95% CI, 0.05–0.60); ctDNA-based: 16.5 versus 5.7 months, HR = 0.19 (95% CI, 0.07–0.55)). Furthermore, longitudinal ctDNA genotyping in the TRIUMPH trial revealed that at least one actionable alteration emerged after disease progression in 16 (62%) of 26 patients, with an enrichment in genes related to the RTK/RAS and PI3K pathways, similar to the situation in CRC with primary resistance. In an exploratory analysis of the DESTINY-CRC01 data, an objective response was observed in patients with or without plasma *RAS* or *PIK3CA* mutations, suggesting the potential of HER2-ADCs to overcome resistance by concurrent genomic alterations [38].

Although T-DXd demonstrated superior efficacy over standard-of-care treatment in previously treated advanced HER2-low breast cancer, no objective response was observed in patients with HER2-low mCRC, defined as IHC 2+ and FISH-negative or IHC 1+ in the DESTINY-CRC01 study [37]. In an observational study, patients with HER2-low mCRC were heterogeneous for HER2-expressing cells, ranging from 5% to 60%, and had a higher proportion of *RAS* mutations and longer OS than did those with HER2-positive mCRC [39]. In addition, PFS after anti-EGFR therapy was also better in those who were HER2-low. These findings suggest that HER2-low mCRC may have a different biological behavior from that of HER2-positive mCRC.

## 6. Future Directions

Although several HER2-targeted strategies have demonstrated promising results in patients with HER2-positive mCRC, their efficacy remains limited. As mentioned previously, preclinical studies using xenografts of HER2-amplified tumor have shown no benefit with monotherapy with anti-HER2 antibodies (pertuzumab) or TKI (lapatinib), while combination therapy with pertuzumab and lapatinib showed anti-tumor activity [11]. However, clinical trials of dual HER2 blockade therapy such as HERACLES-A, MyPathway, and TRIUMPH have demonstrated an ORR of at most 30%. Various resistance mechanisms, including the RTK/RAS and PI3K pathways, may be involved in these modest efficacies, but they have not yet been fully elucidated. Currently, numerous clinical trials for HER2-positive CRC are underway, and a variety of treatment strategies are being developed, including combinations of anti-HER2 antibodies, anti-HER2 antibodies and TKIs, ADCs, and anti-HER2 drugs and immune checkpoint inhibitors (ICIs) (Table 3 and Table 4). These clinical trials include exploratory studies using NGS and ctDNA, which are expected to elucidate more effective anti-HER2 therapies and genetic background in the future.

SWOG S1613, a trastuzumab and pertuzumab trial similar to MyPathway, TAPUR, and TRIUMPH, was a multicenter, randomized, phase II clinical trial that compared the efficacy of cetuximab and irinotecan in patients with *RAS*/*BRAF* wild-type HER2-positive mCRC (NCT03365882). A new HER2-directed antibody, zanidatamab (ZW25), is a bispecific monoclonal antibody that binds to two distinct extracellular epitopes of HER2 and is designed to elicit a cytotoxic T-lymphocyte response and antibody-dependent cell-mediated cytotoxicity against tumor cells overexpressing HER2 [40]. A multicenter, global, phase II study is currently under way that is evaluating the safety and efficacy of zanidatamab in combination with standard chemotherapy in patients with HER2-positive gastrointestinal cancer, including CRC (NCT03929666). In the HERACLES-B and DESTINY-CRC01 trials, treatment using ADCs has been shown to be a promising therapeutic option [33,37]. DESTINIY-CRC2 is a phase II clinical trial evaluating the efficacy and safety of T-DXd at a lower dose (5.4 mg/kg) than the standard dose (6.4 mg/kg) in patients with HER2-positive mCRC. The DASH trial is a phase I/Ib study of T-DXd in combination with ATR inhibition (AZD6738) in advanced solid tumors overexpressing HER2. In addition to T-DXd, several ADCs have been developed, including ZW49 with cytotoxic auristatin conjugated to internalization-enhanced ZW25 (NCT03821233) [41]. Disitamab vedotin (RC48) is a novel HER2-targeted ADC drug developed in China and comprises hertuzumab coupled with auristatin via a cleavable linker. Distamab vedotin has demonstrated safety and potent antitumor activity in a phase I trial in patients with advanced HER2-positive gastric cancer and has been approved in China as second-line treatment for patients with HER2-overexpressing advanced or metastatic gastric/gastroesophageal junction adenocarcinoma [42]. As a novel drug, disitamab vedotin is undergoing clinical trials for various types of cancers, including CRC, mainly in China.

MOUNTAINEER-03 is currently the only phase III trial being conducted in patients with HER2-positive mCRC [43]. This study aims to determine whether combination therapy with tucatinib (HER2-TKI), trastuzumab, and mFOLFOX6 is better than the standard of care for treating patients with HER2-positive mCRC. As the MOUNTAINEER-01 trial yielded very favorable results, further clinical trials assessing the efficacy of novel TKIs, including tucatinib and pyrotinib, for treating HER2-positive mCRC are currently underway. In contrast to tucatinib, which is an orally selective HER2 TKI, pyrotinib is a pan-HER TKI.

ICIs are promising for patients with mCRC with high microsatellite instability and defective DNA mismatch repair. Combination therapy with ICIs and molecularly targeted agents has been attempted for many other tumor types. In an immunocompromised mouse model, anti-PD-1 antibodies significantly improved the antitumor activity of trastuzumab against HER2-positive tumors through enhanced ADCC effect [44]. The KEYNOTE-811 trial of pembrolizumab plus trastuzumab and chemotherapy for patients with HER2-positive gastric cancer demonstrated a statistically significant 22.7% improvement in ORR in the pembrolizumab group compared with the placebo group (77.4% vs. 51.9%, *p* ≥ 0.00006) [45]. Based on these backgrounds, several clinical trials involving patients with HER2-positive CRC have also been designed and conducted using combination treatment with ICIs. Chimeric antigen receptor (CAR)-modified T cell therapy for solid tumors has recently been investigated [46]. VISTA is the first human phase I study to investigate the safety and efficacy of using special immune cells, called HER2 CAR-specific cytotoxic T lymphocytes (HER2-specific CAR T cells), in combination with intra-tumor injection of CAdVEC, an oncolytic adenovirus that is designed to help the immune system, including HER2-specific CAR T cells, react to the tumor.

## 7. Conclusions

In recent years, tissue and ctDNA genomic analysis has demonstrated the utility for identification of HER2-positive mCRC and exploration of resistance mechanisms. A deeper understanding of the molecular biology of HER2-positive mCRC will provide greater insight into the correlations between molecular biology and responsiveness and/or resistance to HER2-targeted therapies and will further accelerate development of therapies for patients with HER2-positive mCRC.

## Figures and Tables

**Figure 1 cancers-15-00183-f001:**
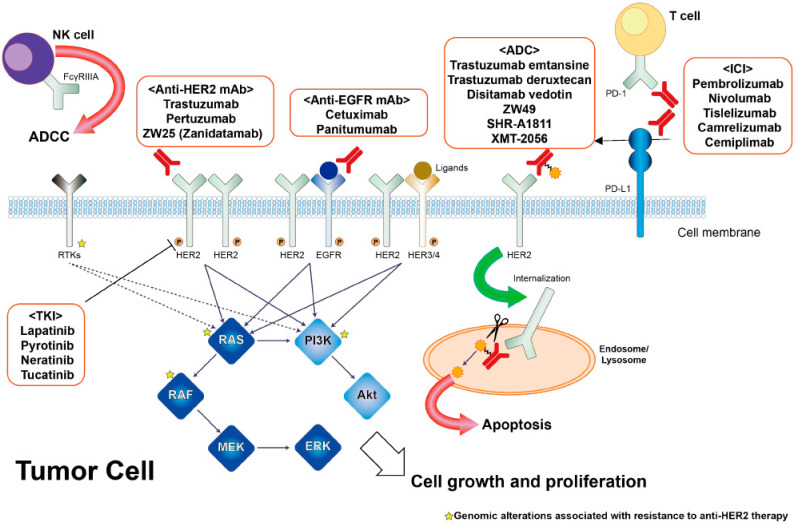
Molecular strategies and resistance mechanism of HER2-targeted treatments. Abbreviations: ADCC, antibody-dependent cellular cytotoxic; ADC, antibody-drug conjugate; ICI, immune checkpoint inhibitor; TKI, tyrosine kinase inhibitor.

**Table 1 cancers-15-00183-t001:** Eligibility criteria of TRIUMPH study.

IHC	IHC Pattern	Classification	Eligibility
Surgical specimen	Complete lateral or circumferential membrane staining with strong intensity and within >10% of tumor cells	Positive IHC (3+)	Eligible (tissue +)
Biopsy specimen	Tumor cells with complete lateral of circumferential membrane staining with strong intensity irrespective of percentage of tumor cells stained		
Surgical specimen	Incomplete lateral or circumferential membrane staining with weak/moderate intensity and within >10% of tumor cells or complete lateral or circumferential membrane staining with strong intensity and within <10% of tumor cells	Equivocal IHC (2+)	Not eligible
Biopsy specimen	Tumor cells with weak to moderate complete circumferential or weak to moderate lateral membrane staining irrespective of percentage of tumor cells stained		
Surgical specimen	Segmental of granular staining in any cellularity	Negative IHC (1+)	Not eligible
Biopsy specimen	Tumor cells with faint or barely perceptible membrane staining irrespective of percentage of tumor cells stained		
Surgical specimen	No staining is observed or incomplete membrane staining with faint/barely perceptible intensity and within ≤10% of tumor cells	Negative IHC (0+)	Not eligible
Biopsy specimen	No staining in any tumor cell		
FISH	*HER2*/*CEP17* ratio		
Surgical or biopsy specimen	≥2	Positive	Eligible (tissue +)
	<2	Negative	Not eligible
Plasma NGS	HER2 status by Gurdant360		
ctDNA	*HER2* amplification (and KRAS/NRAS wild type)	Positive	Eligible (blood +)
	No *HER2* amplification	Negative	Not eligible

Eligible (tissue +): if a patient meets either of the following criteria based on analysis of tumor tissue in central pathological assessment* using the HER2 IHC test and *HER2* FISH test, IHC3+ or FISH-positive (*HER2/CEP17* ratio ≥ 2.0). Eligible (blood +): if a patient has *HER2* amplification (HER2-positive) and wild-type *RAS*^†^ based on the analysis of a blood sample in central assessment* using liquid biopsy. ^†^ Wild-type *RAS* is defined as “Relative clonality^††^, i.e., ≤ 30% in each of *KRAS* codon 12, 13, 59, 61, 117, and 146, and *NRAS* of codon 12, 13, 59, 61, 117, and 146”. ^††^ Relative clonality (%) = % cfDNA of a certain mutation/highest % cfDNA × 100. * The HER2 test of tumor tissue, liquid biopsy using blood samples, and central assessment of the results are performed separately in clinical research. Abbreviations: IHC, immunohistochemistry; FISH, fluorescence in situ hybridization; NGS, next-generation sequencing; cfDNA, cell-free DNA; ctDNA, circulating tumor DNA.

**Table 2 cancers-15-00183-t002:** Efficacy of clinical trials targeting HER2-positive mCRC in which NGS was performed.

Clinical Trial		n	PFS	
HERACLES [29,31]			(weeks)	
qPCR (tissue)	CN ≥ 9.45	18	29	HR 0.67 (95% CI 0.6–0.8), *p* = 0.0001
	CN < 9.45	9	16	
NGS (plasma)	ApCN ≥ 25.82	15	22.5	*p* = 0.0347
	ApCN < 25.82	13	14.8	
TRIUMPH [14]			(months)	
NGS (tissue)	CN ≥ 68.7	9	6.2	HR 0.28 (95% CI 0.11–0.74)
	CN < 68.7	20	2.2	
NGS (plasma)	ApCN ≥ 16.7	13	5.6	HR 0.14 (95% CI 0.05–0.39)
	ApCN < 16.7	16	1.6	
DESTINY-CRC01 [38]			(months)	
NGS (plasma)	ApCN ≥ 30.9	24	10.9	
	ApCN < 30.9	28	4.1	

Abbreviations: NGS, next-generation sequencing; CN, copy number; ApCN, adjusted plasma copy number; PFS, progression-free survival; HR, hazard ratio; CI, confidence interval; mCRC, metastatic colorectal cancer.

**Table 3 cancers-15-00183-t003:** Ongoing trials targeting HER2-positive mCRC.

Phase	NCT Number	Title	Status	Treatment	n	HER2 Status
III	NCT05253651	A Study of Tucatinib With Trastuzumab and mFOLFOX6 Versus Standard of Care Treatment in First-line HER2+ Metastatic Colorectal Cancer (MOUNTAINEER-03)	Recruiting	Tucatinib in combination with Trastuzumab and mFOLFOX6	400	HER2+ disease as determined by a tissue-based assay
II	NCT03365882	S1613, Trastuzumab and Pertuzumab or Cetuximab and Irinotecan Hydrochloride in Treating Patients With Locally Advanced or Metastatic HER2/Neu Amplified Colorectal Cancer That Cannot Be Removed by Surgery	Active, not recruiting	Trastuzumab + Pertuzumab	240	IHC 3+ or IHC 2+ and ISH with HER2/CEP17 ratio ≥2.0
II	NCT03929666	A Safety and Efficacy Study of ZW25 (Zanidatamab) Plus Combination Chemotherapy in HER2-expressing Gastrointestinal Cancers, Including Gastroesophageal Adenocarcinoma, Biliary Tract Cancer, and Colorectal Cancer	Recruiting	ZW25 (zanidatamab) + chemothrapy	362	IHC 3+ or HER2 amplification (based upon central assessment)
II	NCT04380012	A Clinical Study of Pyrotinib in Patients With HER2-positive Advanced Colorectal Cancer	Recruiting	Pyrotinib + Trastuzumab	40	IHC 3+ or 2+ in more than 50% of cells, confirmed by SISH or FISH with HER2/CEP17 ratio ≥ 2.0
II	NCT04744831	Trastuzumab Deruxtecan in Participants With HER2-overexpressing Advanced or Metastatic Colorectal Cancer (DESTINY-CRC02)	Active, not recruiting	T-DXd	122	IHC 3+ or IHC 2+/ISH +
II	NCT05193292	Camrelizumab Combined With Trastuzumab and Chemotherapy in Patients With HER2-positive Advanced Colorectal Cancer	Not yet recruiting	Camrelizumab + Trastuzumab + Chemotherapy	77	HERACLES diagnostic criteria or NGS sequencing of tumor tissue/blood samples showed HER2 amplification.
II	NCT05350917	Study of Tislelizumab Combined With DisitamabVedotin and Pyrotinib Maleate in HER2-positive or Mutated Advanced Colorectal Cancer Who Failed Standard Therapy	Not yet recruiting	Tislelizumab combined with Disitamab Vedotin and Pyrotinib	20	-
II	NCT05333809	Pembrolizumab and Disitamab Vedotin in HER2-expressing Metastatic Colorectal Cancer	Not yet recruiting	Pembrolizumab + Disitamab Vedotin	30	IHC 3+ or IHC 2+
II	NCT05356897	Tucatinib Combined With Trastuzumab and TAS-102 for the Treatment of HER2 Positive Metastatic Colorectal Cancer in Molecularly Selected Patients, 3T Study	Not yet recruiting	Tucatinib combined with Trastuzumab and TAS-102	30	HER2 3+ or IHC 2+/FISH or CISH with signal ratio > 2.2 or gene copy number > 6 or HER2 amplification by NGS
I/II	NCT04278144	A First-in-human Study Using BDC-1001 as a Single Agent and in Combination With Nivolumab in Advanced HER2-Expressing Solid Tumors	Recruiting	BDC-1001+/−Nivolumab	390	-
I	NCT03821233	A Dose Finding Study of ZW49 in Patients With HER2-Positive Cancers	Recruiting	ZW49	174	HER2 high
I	NCT04460456	A Study of SBT6050 Alone and in Combination With PD-1 Inhibitors in Subjects With Advanced HER2 Expressing Solid Tumors	Active, not recruiting	SBT6050 + Pembrolizumab or Cemiplimab	58	IHC 3+ or 2+
I	NCT04704661	Testing the Combination of Two Anti-cancer Drugs, DS-8201a and AZD6738, for The Treatment of Patients With Advanced Solid Tumors Expressing the HER2 Protein or Gene, The DASH Trial	Recruiting	T-DXd and Ceralasertib (AZD6738)	15	IHC 1–3+ or HER2 amplification based on FISH or NGS
I	NCT04513223	A Phase I Study of SHR-A1811 in Patients With Selected HER2 Expressing Tumors	Recruiting	SHR-A1811	114	-
I	NCT05514717	A Study of XMT-2056 in Advanced/Recurrent Solid Tumors That Express HER2	Not yet recruiting	XMT-2056	144	HER2+ will be determined by institutional practice (e.g., IHC, ISH, or NGS)

Abbreviations: IHC, immunohistochemistry; FISH, fluorescence in situ hybridization; NGS, next-generation sequencing; mCRC, metastatic colorectal cancer.

**Table 4 cancers-15-00183-t004:** Ongoing trials using immune cell therapy targeting HER2-positive mCRC.

Phase	NCT Number	Title	Status	Treatment	n	HER2 Status
I	NCT04319757	ACE1702 in Subjects With Advanced or Metastatic HER2-expressing Solid Tumors	Recruiting	ACE1702	36	IHC 3+ or 2+
I	NCT03740256	Binary Oncolytic Adenovirus in Combination With HER2-Specific Autologous CAR VST, Advanced HER2 Positive Solid Tumors (VISTA)	Recruiting	HER2 specific CAR-T cell + CAdVEC	45	IHC 2+ or above
I	NCT04660929	CAR macrophages for the Treatment of HER2-Overexpressing Solid Tumors	Recruiting	CT-0508	18	HER2 positive

Abbreviations: IHC, immunohistochemistry; CAR, chimeric antigen receptor; mCRC, metastatic colorectal cancer.

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
