# Peer review of "Molecular Basis of HER2-Targeted Therapy for HER2-Positive Colorectal Cancer"

_cancers, 2022, doi:10.3390/cancers15010183_

Round 1

Reviewer 1 Report

Authors performed a well-presented study.

I would like to recommend to add more information about HER2 kinase and receptor tyrosine kinases.

HER2 has no specific ligand and generally generate dimers with EGFR. Authors could also add some information about EGFR-HER2 dual targets in colorectal cancer.

Reviewer 2 Report

In this article the “Molecular basis of HER2-targeted therapy for HER2-positive colorectal cancer” the authors have highlighted the role and updates on HER2-targeted therapy. The manuscript is well written and have provided the comprehensive insights however below are the minor observations that need to be addressed.

·         It would be decent if the authors could add some pictorial representations in supplement to their manuscript text.  

·         Abbreviations details is missing someplace at first use.

·         Conclusion is well written; we suggest to elaborate conclusion more in the front of future directions and urgent open gaps that need to be addressed.

Reviewer 3 Report

work well set up and correct from a laboratory and clinical point of view. We should specify more and why even better recognizing Her-2 receptors we have not been able to have real therapeutic advantages. It is not clear why immunotherapy associations can be useful. The limits should be defined more and the poor successes given proper weight. I would say that a little more criticism is needed.
